# Insights into Electron Transport in a Ferroelectric Tunnel Junction

**DOI:** 10.3390/nano12101682

**Published:** 2022-05-14

**Authors:** Titus Sandu, Catalin Tibeica, Rodica Plugaru, Oana Nedelcu, Neculai Plugaru

**Affiliations:** National Institute for Research and Development in Microtechnologies-IMT, 126A, Erou Iancu Nicolae Street, 077190 Voluntari Ilfov, Romania; catalin.tibeica@imt.ro (C.T.); rodica.plugaru@imt.ro (R.P.); oana.nedelcu@imt.ro (O.N.)

**Keywords:** ferroelectric tunnel junction, electron transport, non-equilibrium Green function, resonance states, empirical tight-binding

## Abstract

The success of a ferroelectric tunnel junction (FTJ) depends on the asymmetry of electron tunneling as given by the tunneling electroresistance (TER) effect. This characteristic is mainly assessed considering three transport mechanisms: direct tunneling, thermionic emission, and Fowler-Nordheim tunneling. Here, by analyzing the effect of temperature on TER, we show that taking into account only these mechanisms may not be enough in order to fully characterize the performance of FTJ devices. We approach the electron tunneling in FTJ with the non-equilibrium Green function (NEGF) method, which is able to overcome the limitations affecting the three mechanisms mentioned above. We bring evidence that the performance of FTJs is also affected by temperature–in a non-trivial way–via resonance (Gamow-Siegert) states, which are present in the electron transmission probability and are usually situated above the barrier. Although the NEGF technique does not provide direct access to the wavefunctions, we show that, for single-band transport, one can find the wavefunction at any given energy and in particular at resonant energies in the system.

## 1. Introduction

The asymmetric ferroelectric tunnel junction (FTJ), a thin ferroelectric (FE) layer sandwiched between two dissimilar electrodes–or with different interfaces in the case of the same electrode material–is a promising electron device for many applications such as low power memories [1] or neuromorphic computing [2]. The salient mechanism of FTJ is based on the tunneling electroresistance (TER) effect, i.e., a change in the electrical resistivity when the ferroelectric polarization is reversed under an external electric field [3,4]. In other words, the electrical resistance of FTJ switches from a high conduction (ON) state to a low conduction (OFF) state or vice-versa when a voltage pulse is applied. Ferroelectrics like BaTiO_3_, PbZr_0.2_Ti_0.8_O_3_, BiFeO_3_ [5,6,7], as well as high-k dielectrics like HfO_2_ and Hf_0.5_Zr_0.5_O_2_ [8,9] have been shown to work as FTJs. The latter are particularly attractive since they are compatible with the CMOS technology. The TER effect can be characterized by the TER ratio
(1)$$\mathrm{TER}=\frac{\left(J_{ON}-J_{OFF}\right)}{J_{OFF}},$$
where *J_ON/OFF_* are the current densities in the two different states. The higher the TER ratio and *J_ON_*, the better the FTJ performance. The extent of the barrier modulation depends on numerous factors, such as: (i) the thickness and spontaneous polarization of the ferroelectric film, (ii) the bias voltage, (iii) the differences between work functions and screening lengths of the electrodes, and (iv) the built-in field and screening of the polarization charge, as well as on the variation of barrier thickness due to piezoelectricity [10]. In some FTJ implementations, an ultrathin dielectric (DE) layer is also introduced, as shown in Figure 1, in order to ensure an additional way to tune the FTJ performance.

To understand, describe, and model the TER effect, the ab-initio methods may provide an accurate image of the physics governing FTJ systems including the geometry, polarization state, electronic structure [11], and transport properties [12,13]. Nevertheless, ab-initio methods require large computational resources, which may prevent their intensive use in the early stages of FTJs design, when there is a requirement for a fast scan in parameter spaces to obtain a device with required basic functionalities. In the search for optimized devices, the semi-empirical methods are fast and, despite all simplifications, are extremely useful for the treatment of the charge transport as long as they are fed with appropriate physical parameters [14,15]. These semi-empirical approaches are based on the non-equilibrium Green function (NEGF) method that can calculate exactly, in principle, the tunneling current and the I–V characteristic of a FTJ [15].

In semi-empirical methods based on NEGF, the parameters can be adjusted to match the experimental data [14]. In many cases, in order to characterize the electron transport and interpret the I–V curves, various models of transport mechanisms, involving direct tunneling, thermionic emission, and Fowler-Nordheim tunneling, are used [16]. Direct tunneling presumes low energy for incident electrons such that the tunneling probability does not depend on the profile of the barrier, which is assumed to be rectangular on average in the case of Simmons formula [17,18] or trapezoidal in the case of Brinkman et al. formula [18,19]. The thermionic emission is that part of the current carried over the top of the barrier by thermally activated charge carriers [20]. The last mechanism, the Fowler-Nordheim tunneling, is used for triangular barrier profiles, which are encountered at finite applied bias voltages [21]. The treatment of the above mechanisms is based on several approximations for electron transport. The Simmons and Brinkman et al. as well as the Fowler-Nordheim formulae are based on the semi-classical Wentzel–Kramers–Brillouin (WKB) approximation [17,21], while thermionic emission tacitly assumes a homogeneous transmission probability of one for all electrons with energy above the top of the barrier [20]. In this work, by the exact treatment of the tunneling within the NEGF method, we show that the above models (describing direct tunneling, thermionic emission, and Fowler-Nordheim tunneling) are not accurate enough for performance characterization of FTJs. These models cannot capture resonant features emerging in FTJ structures that might play an important role in carrier transport and ultimately in the performances of the device. The electric current of an impinging electron at a given energy is determined by the electron transmission probability at that energy weighted (multiplied) by its Fermi-Dirac distribution function. Hence, even though they seem unlikely to play a role in the tunneling due to the relatively simple structure of the barriers, the resonances can contribute significantly to the electric current since their contribution to transmission probability may offset the temperature-dependent Fermi-Dirac factor. In the following, we will show that the resonances emerging close or above the top of the barriers play a decisive role for transport at room temperature, a fact that is not captured by the models previously mentioned. Moreover, resonances may appear in FTJs with composite (both a ferroelectric and a dielectric) barriers. The NEGF method can also accurately describe these type of heterostructures. Generally, the NEGF approach cannot provide full access to the wavefunctions, especially for multi-band transport [22]. However, for a single-band transport, we show that this is not the case; the eigenvectors of the spectral functions are just the wavefunctions of the system. At this stage, we are able to identify the Green function of the device as the discrete version of the outgoing Green function, which is used in diverse scattering problems like nuclear reactions or electron transport in nanostructures [23,24]. The advantage of such correspondence is that the NEGF of the device and the transmission probability can be expanded as a sum of resonances or Gamow-Siegert states [23,24,25,26]. Thus, we are able to sort out the resonances that count for the electron transport in FTJs.

The paper is organized as follows: The next section deals with the theoretical background of the calculations of applied voltage induced electrostatic potential across the FTJ, of transport quantities like I–V curve and conductance by NEGF method, and the evaluation of wavefunctions and resonance states from NEGF calculations. Section 3 presents numerical results regarding the effect of temperature on electric conductance and TER ratio for several practical cases of simple and composite BaTiO_3_-based FTJs. In addition, the wavefunctions and resonance states for such FTJ structures are analyzed. Section 4 summarizes the conclusions of this work, and lastly, in Appendix A we present the steps to obtain a discrete tight-binding Hamiltonian from the BenDaniel and Duke Hamiltonian [22].

## 2. Theoretical Background

### 2.1. The Profile of Potential Barrier

In semi-empirical methods, the calculation of tunneling current and electric conductance is performed by simultaneously solving the electrostatic and transport problems. They are intricately interrelated since the charge density in Poisson equation is calculated self-consistently from the NEGFs [22]. However, when one deals with free carriers, only in the metallic contacts they can decouple each other. Thus, let us first deal with the electrostatics and the profile of the potential barrier. We assume that the dielectric and ferroelectric are located at *x* between *−t_DE_* and 0 and *x* between 0 and *t_FE_*, respectively, while the metallic contact ML is at *x* < −*t_DE_* and the metallic contact MR is at *x* > *t_FE_*, as illustrated in Figure 1. A widely used and good approximation of electrostatics in metals is the Thomas-Fermi approximation [14,17]. Within this framework, the electrical fields in ML and MR are given by:(2)$$E_{ML}\left(x\right)=\tau_{S}e^{\left(x+t_{DE}\right)/\lambda_{1}}/\left(\varepsilon_{1}\varepsilon_{0}\right),$$
(3)$$E_{MR}\left(x\right)=\tau_{S}e^{-\left(x-t_{FE}\right)/\lambda_{2}}/\left(\varepsilon_{2}\varepsilon_{0}\right).$$

In Equations (2) and (3) *τ_S_* is the screening charge at ML/dielectric and MR/ferroelectric interfaces, *λ*_1_*(ε*_1_*)* and *λ*_2_*(ε*_2_*)* are the Thomas-Fermi screening lengths (dielectric constants) of ML and MR, respectively, and *ε_0_* is the vacuum permittivity. In the following, we denote by *ε_FE_*, *E_FE_*, and *P* the dielectric constant, the electric field, and the intrinsic polarization, respectively, of the ferroelectric, and by *E_DE_* and *ε_DE_* the electric field and the dielectric constant of the dielectric, respectively. The electrostatic equations are in fact the continuity of the normal component of the electric induction *D* at both ferroelectric-MR and ML-dielectric interfaces.
(4)$$\tau_{S}=\varepsilon_{FE}\varepsilon_{0}E_{FE}+P=\varepsilon_{DE}\varepsilon_{0}E_{DE}.$$

Additionally, the bias voltage, *V*, across the structure obeys the equation:(5)$$\frac{\tau_{S}\lambda_{1}}{\varepsilon_{1}\varepsilon_{0}}+\frac{\tau_{S}\lambda_{2}}{\varepsilon_{2}\varepsilon_{0}}+E_{FE}t_{FE}+E_{DE}t_{DE}+V+V_{BI}=0,$$
where *V_BI_* is the built-in voltage due to mismatch of the conduction band discontinuities and of Fermi energies *E_FL_* (*E_FR_*) of ML (MR).
(6)$$V_{BI}=\left(\varphi_{2}+\varphi_{C}-E_{FR}-\varphi_{1}+E_{FL}\right)/e.$$

In (6), *e* is the elementary electric charge, *φ*_1_ is the band discontinuity at the first interface between ML and the dielectric, *φ*_2_ is the band discontinuity at the second interface between the ferroelectric and MR, and *φ_C_* is the band discontinuity at the interface between the dielectric and ferroelectric when the ferroelectric is unpolarized. Eliminating *E_FE_* and *E_DE_* from (4) and (5), we obtain the screening charge *τ_S_*
(7)$$\tau_{S}=\frac{-\left(V+V_{BI}\right)+t_{FE}P/\left(\varepsilon_{0}\varepsilon_{FE}\right)}{\frac{\lambda_{1}}{\varepsilon_{0}\varepsilon_{1}}+\frac{\lambda_{2}}{\varepsilon_{0}\varepsilon_{2}}+\frac{t_{FE}}{\varepsilon_{0}\varepsilon_{FE}}+\frac{t_{DE}}{\varepsilon_{0}\varepsilon_{DE}}}.$$

Knowing that the electric field is homogeneous in both the dielectric and the ferroelectric, now it is straightforward to obtain the electric potential. Then, the tunneling barrier profile for electrons has the following form:(8)$$U\left(x\right)=\{\begin{array}{l} e\frac{\tau_{s}\lambda_{1}}{\varepsilon_{1}\varepsilon_{0}}exp\left[\left(x+t_{DE}\right)/\lambda_{1}\right],\;x<-t_{DE}\\ e\tau_{s}\left(\frac{\lambda_{1}}{\varepsilon_{1}\varepsilon_{0}}+\frac{t_{DE}+x}{\varepsilon_{DE}\varepsilon_{0}}\right)+\varphi_{1},\;-t_{DE}\leq x<0\\ e\tau_{s}\left(\frac{\lambda_{1}}{\varepsilon_{1}\varepsilon_{0}}+\frac{t_{DE}}{\varepsilon_{DE}\varepsilon_{0}}\right)+e\frac{\tau_{s}-P}{\varepsilon_{FE}\varepsilon_{0}}x+\varphi_{1}-\varphi_{C},\;0\leq x\leq t_{FE}\\ -e\left(V+V_{BI}+\frac{\tau_{s}\lambda_{2}}{\varepsilon_{2}\varepsilon_{0}}exp\left[-\left(x-t_{FE}\right)/\lambda_{2}\right]\right)+\varphi_{1}-\varphi_{C}-\varphi_{2},\;x>t_{FE} \end{array}$$

In Equation (8), *P* is considered positive when points from ML to MR, and negative when it points the other way around.

### 2.2. Transport by NEGF

Transport properties like electric current density and conductance can be calculated by solving the Schrödinger equation with scattering boundary conditions, i.e., either an incoming wave from the left or from the right. For a single band, the equation is merely 1D with a BenDaniel and Duke Hamiltonian, where the effective mass of electrons can vary across the structure [22]:(9)$$H=-\frac{\hbar^{2}}{2}\frac{d}{dx}\left(\frac{1}{m^{*}\left(x\right)}\frac{d}{dx}\right)+\frac{\hbar^{2}k^{2}}{2m^{*}\left(x\right)}+U\left(x\right).$$

In Equation (9), it is assumed that the energy band is parabolic with an isotropic effective mass in each layer of the heterostructure, ***k*** is the transverse wavevector (parallel to each interface), *m** is the effective mass, and *U(x)* the potential energy given by (8). Equation (9) can be discretized into a 1D problem in a tight-binding representation [22]. The details are presented in Appendix A. In general, in the tight binding representation, the Hamiltonian has a block matrix form, or more precisely, a block tridiagonal form, as in Equation (10):(10)$$H=\left(\begin{array}{ccc} H_{L} & V_{LD} & 0\\ V_{LD}^{†} & H_{D} & V_{RD}^{†}\\ 0 & V_{RD} & H_{R} \end{array}\right).$$

For a single band model, the block tridiagonal matrix form of H becomes a simple tridiagonal matrix. In this format, one can easily see that the Hamiltonian has three parts: the semi-infinite left (L) and right (R) metallic electrodes and the device (D) that is defined by *H_D_*, an *n_D_* × *n_D_* matrix. Both electrodes act as reservoirs, hence they have well-defined chemical potentials and temperatures. We can define the retarded Green function (*G^R^*) of the system at energy *E* as the inverse matrix of [(*E* + *iη*) − *H*], where *η* = 0+. Similarly, the advanced Green function *G^A^* is the inverse of [(*E* − *iη*) − *H*], i.e.,
(11)$$G^{R,A}\left(E\right)={\left[\left(E\;\pm \;i\eta \right)-\;H\right]}^{-1}.$$

In the NEGF method, one eliminates the degrees of freedom of contacts by introducing self-energies into the projected Green functions of the device (D).
(12)$$G_{D}^{R,A}\left(E\right)={\left[\left(E\;\pm \;i\eta \right)-{\;H}_{D}-\Sigma_{B}^{R,A}\left(E\right)\right]}^{-1}.$$

The self-energy $\Sigma_{B}^{R,A}\left(E\right)=\Sigma_{L}^{R,A}\left(E\right)+\Sigma_{R}^{R,A}\left(E\right)$ has two components due to the coupling to the left and right contacts. It replaces the boundary conditions that otherwise would be fulfilled by the construction of a Green function in the device region. The self-energy due to the left contact has the following expression:(13)$$\Sigma_{L}^{R,A}\left(E\right)=V_{LD}g_{L}^{R,A}\left(E\right)V_{LD}^{†},$$
where $g_{L}^{R,A}\left(E\right)={\left[\left(E\;\pm \;i\eta \right)-{\;H}_{L}^{0}\right]}^{-1}$ is the Green function of the semi-infinite left contact. Additionally, $\Sigma_{R}^{R,A}\left(E\right)$ has a similar expression. Here we notice that due to the fact that we deal with a nearest-neighbor tight-binding Hamiltonian, the self-energies $\Sigma_{L}^{R}$ and $\Sigma_{R}^{R}$ are *n_D_* × *n_D_* matrices with just one non-zero element, the (1,1) element for $\Sigma_{L}^{R}$ and (*n_D_*,*n_D_*) for $\Sigma_{R}^{R}$. Thus, the retarded Green function is:(14)$$G_{D}^{R}={\left(\begin{array}{ccccc} E+i\eta -D_{1}+\sigma_{L}^{R}\left(E\right) & -t_{12} & 0 & \cdots & 0\\ -t_{21} & E+i\eta -D_{2} & \ddots & \ddots & \vdots \\ 0 & \ddots & \ddots & -t_{n_{D}-2,n_{D}-1} & 0\\ \vdots & \ddots & -t_{n_{D}-1,n_{D}-2} & E+i\eta -D_{n_{D}-1} & -t_{n_{D}-1,n_{D}}\\ 0 & \cdots & 0 & -t_{n_{D},n_{D}-1} & E+i\eta -D_{n_{D}}+\sigma_{R}^{R}\left(E\right) \end{array}\right)}^{-1}$$

The poles of the device Green function are no longer real–an attribute of an open quantum system. Starting from Equation (11), the calculation of $\sigma_{L}^{R}\left(E\right)$ involves the calculation of matrix element (0,0) of $g_{L}^{R}\left(E\right)$, which is the surface Green function of the left contact. Bearing in mind that *D_L_* and *t_L_* are the tight-binding parameters of the left contact and defining a longitudinal wavevector *k_l_*, the energy can be parameterized according to the single-band dispersion relation for the left contact $E=D_{L}-2t_{L}\cos\left(k_{l}\Delta \right)$. One can find that the matrix element (0,0) of $g_{L}^{R}\left(E\right)$ is $-e^{ik_{l}\Delta }/t_{L}$ and $\sigma_{L}^{R}\left(E\right)=-t_{L}e^{ik_{l}\Delta }$ [14,22,27]. The expression of $g_{L}^{R}\left(E\right)$ is calculated with outgoing boundary condition [27], hence the self-energy is for this kind of boundary condition.

One can further define: (a) the spectral function $A=i\left(G^{r}-G^{a}\right)$, which also has a matrix form–whose diagonal is just the local density of states up to a 2π factor–and (b) the broadening function due to the coupling to the left and right contacts $\Gamma_{L,R}\left(E\right)=i\left(\Sigma_{L,R}\left(E\right)-\Sigma_{L,R}^{†}\left(E\right)\right)$.

Using the projection operator on the device *D,* one can show that the projection of the full spectral function *A* on the device space is [28].
(15)$$A_{D}\left(E\right)=i\left(G_{D}^{R}\left(E\right)-G_{D}^{A}\left(E\right)\right)=G_{D}^{R}\left(E\right)\left(\Gamma_{L}\left(E\right)+\Gamma_{R}\left(E\right)\right)G_{D}^{A}\left(E\right).$$

Moreover, one can further show that partial spectral densities
(16)$$A_{L,R}\left(E\right)=G_{D}^{R}\left(E\right)\Gamma_{L,R}\left(E\right)G_{D}^{A}\left(E\right)$$
are spectral densities due to incident Bloch waves that come from the left (L) and from the right (R), respectively [28]. Thus, $G_{D}^{R}\left(E\right)$ contains information about both solutions of the scattering problem. Furthermore, it can be shown that the current flow from an incident Bloch wave that comes from the left electrode into the right electrode is
(17)$$j\left(E\right)=\frac{2e}{h}Tr\left[G_{D}^{R}\left(E\right)\Gamma_{L}\left(E\right)G_{D}^{A}\left(E\right)\Gamma_{R}\left(E\right)\right]$$

$G_{D}^{R}\left(E\right)\Gamma_{L}\left(E\right)G_{D}^{A}\left(E\right)\Gamma_{R}\left(E\right)$ is just the matrix of transmission probability T(E) that has a schematic representation in Figure 2. From Equation (17), the expression of the total current takes the form of the Landauer-Büttiker formula [15,22]:(18)$$J=\frac{2e}{h}\int^{\;}Tr\left(T\left(E\right)\right)\left(f_{L}\left(E\right)-f_{R}\left(E\right)\right)dE.$$

In Equation (18), the *Tr ()* operation includes both the trace over the T(E) matrix and the integration over the transverse wavevector ***k***. The functions *f_L,R_* are the Fermi-Dirac distribution functions of the *L* and *R* electrodes. Performing the trace operation only over the *T* matrix, we obtain the transmission probability coefficient $\tilde{t}$, and the Landauer-Büttiker formula becomes:(19)$$J=\frac{e}{2\pi^{2}h}\int_{-\infty }^{\infty }\int_{-\infty }^{\infty }d^{2}k\int_{0}^{\infty }\tilde{t}\left(k,E\right)\left(f_{L}\left(E\right)-f_{R}\left(E\right)\right)dE.$$

In addition, in the linear regime (small bias voltages V) and at temperature of 0 K, we obtain the Landauer conductance formula [15]
(20)$$G=\frac{2e^{2}}{h}\int_{-\infty }^{\infty }\int_{-\infty }^{\infty }\frac{d^{2}k}{{\left(2\pi \right)}^{2}}\tilde{t}\left(k,E_{F}\right)$$

### 2.3. Retrieving the Wavefunction from the Spectral Function. Resonance States

Let us consider the spectral function $A_{L}\left(E\right)=G_{D}^{R}\left(E\right)\Gamma_{L}\left(E\right)G_{D}^{A}\left(E\right)$. It signifies the projected spectral function on the device for incident waves coming from the left. As it was pointed out in Ref. [28], in general the eigenvectors of *A_L_* cannot be identified with the wavefunction in the device region since there are several eigenvectors of *A_L_* with non-zero eigenvalues. This is rather obvious in the tight-binding representation, but for the multi-band problem. For a single-band problem, however, this is not the case. *A_L_* has just one non-zero eigenvalue. Its corresponding eigenvector is just the function that is proportional to the wavefunction in the device region. This statement can be proven directly. The matrix $\Gamma_{L}\left(E\right)$ has just one element different from 0, like its corresponding self-energies $\Sigma_{L}^{R,A}\left(E\right)$
(21)$$\Gamma_{L}\left(E\right)=\left(\begin{array}{ccc} \gamma_{L}\left(E\right) & 0 & \cdots \\ 0 & 0 & \\ \vdots & & \ddots \end{array}\right).$$

Denoting by $G_{D,i,j}\left(E\right)$ the matrix elements of $G_{D}^{R}\left(E\right)$ and by $A_{L,i,j}\left(E\right)$ the matrix elements of $A_{L}\left(E\right)$, it is easy to check that $A_{L,i,j}\left(E\right)=\gamma_{L}\left(E\right)G_{D,i,1}\left(E\right)G_{D,j,1}^{*}\left(E\right)$, where the *** means complex conjugation. One can further see that the *n_D_*-dimensional vector
(22)$$v_{L}=\left(\begin{array}{ccc} G_{D,1,1}\left(E\right) & G_{D,2,1}\left(E\right) & \begin{array}{cc} \cdots & G_{D,n_{D},1}\left(E\right) \end{array} \end{array}\right)$$
is an eigenvector of $A_{L}\left(E\right)$ with the eigenvalue
(23)$$\lambda_{L}=\gamma_{L}\left(E\right)\sum_{i=1}^{n_{D}}{\left|G_{D,1,i}\left(E\right)\right|}^{2}=Tr\left(A_{L}\left(E\right)\right).$$

The fact that $\lambda_{L}=Tr\left(A_{L}\left(E\right)\right)$ ensures that *λ_L_* is the only non-zero eigenvalue of $A_{L}\left(E\right)$. In a bra and ket notation, we thus write $A_{L}\left(E\right)$ as
(24)$$A_{L}\left(E\right)=\gamma_{L}\left(E\right)|v_{L}\rangle \langle v_{L}|$$
since the norm of $v_{L}$ is just $\sqrt{\lambda_{L}/\gamma_{L}}$. Equations (22) and (24) of $v_{L}$ and $A_{L}\left(E\right)$ guarantee that $v_{L}$ is proportional to the solution of Schrödinger equation for incoming waves from the left projected on the device space:(25)$${|\Psi_{L}\rangle }_{D}=\sqrt{\frac{\gamma_{L}}{2\pi }}v_{L}$$

Similar calculations can be performed for $A_{R}\left(E\right)$, explicitly, the matrix form is $A_{R,i,j}\left(E\right)=\gamma_{R}\left(E\right)G_{D,i,N}\left(E\right)G_{D,j,N}^{*}\left(E\right)$ with the eigenvector
(26)$$v_{R}=\left(\begin{array}{ccc} G_{D,1,n_{D}}\left(E\right) & G_{D,2,n_{D}}\left(E\right) & \begin{array}{cc} \cdots & G_{D,n_{D},n_{D}}\left(E\right) \end{array} \end{array}\right),$$
the eigenvalue
(27)$$\lambda_{R}=\gamma_{R}\left(E\right)\sum_{i=1}^{n_{D}}{\left|G_{D,1,n_{D}}\left(E\right)\right|}^{2}=Tr\left(A_{R}\left(E\right)\right),$$
and the simple bra and ket form
(28)$$A_{R}\left(E\right)=\gamma_{R}\left(E\right)|v_{R}\rangle \langle v_{R}|.$$

Also Equations (26) and (28) of $v_{R}$ and $A_{R}\left(E\right)$ guarantee that $v_{R}$ is proportional to the solution of Schrödinger equation for incoming waves from the right, projected on the device space:(29)$${|\Psi_{R}\rangle }_{D}=\sqrt{\frac{\gamma_{R}}{2\pi }}v_{R}$$

The total spectral function $A_{D}\left(E\right)$ is the sum of $A_{L}\left(E\right)$ and $A_{R}\left(E\right)$, hence its range is spanned by $v_{L}$ and $v_{R}$ with two eigenvalues *λ_1_* and *λ_2_*. They obey the following equation: $\lambda_{1}+\lambda_{2}=\lambda_{L}+\lambda_{R}$. Also it is easy to find that the squared modulus of the overlap between ${|\Psi_{L}\rangle }_{D}$ and ${|\Psi_{R}\rangle }_{D}$ is
(30)|〈D ΨL|ΨR〉D|2=(λL λR−λ1λ2)/4π2

From the analysis we are going to perform in the next section on realistic examples, we will see that for energies in the direct tunneling regime there are two distinct solutions with low overlap. In this case, *λ*_1_ and *λ*_2_ are close to *λ_L_* and λ*_R_*. In the opposite case, at resonance, one of the two eigenvalues *λ*_1_ or *λ*_2_ is much larger than the other, hence the overlap is large. In a similar manner, we can obtain the explicit expression of the transmission probability coefficient in terms of the Green function
(31)$$\tilde{t}\left(E\right)=\gamma_{L}\left(E\right)\gamma_{R}\left(E\right){\left|G_{D,1,n_{D}}\right|}^{2}.$$

Equation (31) has previously been deduced using an iterative procedure to calculate the Green function [22,29]. It additionally shows that the transmission probability coefficient has the spectral properties of the Green function. There is a large body of work in which the Green function $\;G_{D}^{R}\left(E\right)$ is set in a meaningful representation. Such a representation is given by the expansion of the Green function in resonance (Siegert-Gamow) states [23,24], which lead to the Breit-Wigner formula for resonances in transmission. Here we will outline a few results about resonant states representation that are connected with our results discussed above. The expansion of the Green function in resonance states is the sum over these states plus a background, and it may take the following form:(32)$$G_{D}^{R}\left(x,x’,E\right)=\sum_{n=1}^{N}\frac{u_{n}\left(x\right)u_{n}\left(x’\right)}{E-E_{n}}+B\left(E\right).$$
where $u_{n}\left(x\right)$ is a resonant state that satisfies the Schrödinger equation
(33)$$H_{D}u_{n}\left(x\right)=E_{n}u_{n}\left(x\right)$$
with outgoing boundary conditions at the device boundary. Since these boundary conditions are not hermitian, the eigenvalues are rather complex, i.e., $E_{n}=E_{rn}-i\Gamma_{n}/2$. Resonant states come into pairs with a negative and positive imaginary part [23,25,26]. Now, it is rather obvious that for energy *E* far from any resonant energy *E_rn_*, the solution of the Schrödinger equation for incoming waves from the left is different from the solution of the Schrödinger equation for incoming waves from the right due to different mixing of the tails of various resonances. However, for energy *E* near a resonant energy *E_rn_*, both solutions from the left and from the right are similar since the dominant term is that given by $u_{n}\left(x\right)$. Finally, it is quite straightforward to notice by using Equation (29) that the transmission probability coefficient has a multi-resonance Breit-Wigner-like form [24]
(34)$$\tilde{t}\left(E\right)=\sum_{n=1}^{N}T_{n}\left(E\right)+\sum_{n<m}^{N}T_{nm}\left(E\right)+R\left(E\right).$$

The term
(35)$$T_{n}\left(E\right)=\frac{A_{n}}{{\left(E-E_{rn}\right)}^{2}+{\left(\Gamma_{n}/2\right)}^{2}}$$
is the Breit-Wigner expression, $T_{nm}\left(E\right)$ of the interference term between resonances, and R(E) is the term resulting from the background.

## 3. Numerical Analysis of Tunneling in Relevant FTJs

### 3.1. Temperature Influence on Conductance and TER Ratio 

Numerically, we calculated all quantities defined in the previous section using the NEGF formalism. The calculation of I–V characteristics is performed with Equation (19), which is able to reproduce the non-linear regime for large bias voltages and finite temperatures. Equation (20) describes the linear regime at 0 K and is often used (especially in ab-initio calculations, where a full I–V curve is costly) in the calculation of the TER ratio defined by Equation (1). In the following we shall analyze a few practical FTJ structures.

#### 3.1.1. Pt/BaTiO_3_/SrRuO_3_ FTJ

The first system to deal with is that of a BaTiO_3_ ferroelectric barrier on top of a SrRuO_3_ substrate which acts as the right contact layer in Figure 1, and Pt is the left contact to the barrier. The physical parameters are: *λ*_1_ = 0.45 Ǻ, *λ*_2_ = 0.75 Ǻ, *ε*_1_ = 2, *ε*_2_ = 8.45, *ε*_FE_ = 125, *E_FL_* = *E_FR_* = 3 eV, *φ*_1_ = *φ*_2_ = 3.6 eV, *P* = 16 μC/cm^2^. The effective masses are: 5 m_0_ for SrRuO_3_, 2 m_0_ for BaTiO_3_, and m_0_ for Pt, where m_0_ is the free electron mass [30]. The Fermi energy is set to *E_F_* = *E_FL_* = *E_FR_* = 3 eV. One should note that the electron effective masses are different in the three layers of the structure, hence the calculations need full numerical integration over transverse *k* wavevector in Equations (19) and (20). In Figure 3 we show the potential profiles of a 2 nm thick BaTiO_3_ barrier for both directions of polarization. The transmission probability coefficients, also depicted in the right of Figure 3, exhibit resonances above the very top of the barriers. The resonances are associated with the peaks in the transmission spectra and if they are well resolved then they obey Equation (35). In Figure 4a, we plot the conductance of the system, and in Figure 4b the TER ratio at 0 K and 300 K.

The calculation of the conductance at 300 K, see Figure 4a, was performed with an applied bias of 0.0001 V, which is low enough to satisfy the linear regime conditions. Both the conductance and TER ratio behave differently at 300 K with respect to 0 K. At 300 K we observe two regimes depending on the ferroelectric thickness–one regime up to 2.5 nm and another one beyond this value. The difference between the two regimes is explained in Figure 5. At 0 K, the channels open to electron flow are those up to the Fermi energy, represented by dashed line. These channels are those of direct tunneling such that the Simmons and Brinkman formulae are appropriate [17,18,19]. For the barrier of 1.6 nm thickness (Figure 5a), the transport occurs around Fermi energy even at room temperature (300 K), hence the similar behavior of the conductance at 300 K with respect to 0 K, see Figure 4a. There is a small contribution to the transmitted current from the resonance states, but this contribution is orders of magnitude smaller. However, for thicker barriers like the one shown in Figure 5b, things may change. First, the direct tunneling current is much smaller since it has an exponential dependence on the barrier thickness. Second, the resonance levels are spaced much closer, hence their contribution to transmission increases considerably. Even if their occupancy–given by the Fermi-Dirac function–might be small, it is offset by the large transmission probability. In Figure 5b it can easily be seen that the contribution from the resonance states is overwhelmingly larger than the contribution from the states near the Fermi energy. Moreover, electrons with energies closer to the barrier top encounter a triangular barrier profile and so they may still reach a resonance state; this mechanism cannot be described within the semiclassical WKB theory of Fowler-Nordheim [21]. A similar behavior of the TER ratio at room temperature was also observed in experimental data [31]; a degradation of TER was even found when increasing the ferroelectric thickness. However, performing their analysis based on Brinkman model at finite bias voltage, the authors attributed the TER degradation to the rather high levels of noise in the measurements.

#### 3.1.2. Pt/ SrTiO_3_/BaTiO_3_/SrRuO_3_ Composite Barrier FTJ

The second system we analyze is a composite FTJ, where a dielectric barrier (SrTiO_3_) is added besides the BaTiO_3_ ferroelectric barrier. The electrodes are Pt (left) and SrRuO_3_ (right). The role of the dielectric layer is to increase the asymmetry of the system, hence it is expected a higher TER ratio [32]. Physical parameters are slightly changed with respect to the previous case [32,33]: *λ*_1_ = 0.45 Ǻ, *λ*_2_ = 0.8 Ǻ, *ε*_1_ = 2, *ε*_2_ = 8.45, *ε*_FE_ = 90, *ε*_DE_ = 90, *E_FL_* = *E_FR_* = 3 eV, *φ*_1_ = *φ*_2_ = 3.6 eV, *φ*_C_ = 0 eV, *P* = 20 μC/cm^2^. The effective masses are: 5 m_0_ for SrRuO_3_, 2 m_0_ for BaTiO_3_ and SrTiO_3_, and m_0_ for Pt. In the following, BaTiO_3_ thickness is fixed to 2.4 nm and we varied the SrTiO_3_ thickness between 0.5 and 3 nm. The results of the calculations are presented in Figure 6.

At 0 K, the conductance for both polarization directions shows an exponential dependence on the dielectric thickness. At room temperature, on the other hand, it appears that this exponential dependence is no longer valid, as revealed by the TER ratio. In this case, a decrease in TER takes place when the dielectric thickness increases (Figure 6b). The explanation of TER degradation is provided by the results presented in Figure 7. One may observe that the electric charges are mainly transported through quantum states that are close or above the barrier top, the contribution of the states near the Fermi energy being negligible. The total barrier thickness is 3 nm at least, a value at which the resonance states start to play a significant role. As the barrier thickness increases, the contribution of the resonance states is enhanced. Nevertheless, those resonances located above the barrier (see the F-D weighted curves in Figure 7 right panel) become less sensitive to the barrier profile, and hence the decline in the TER ratio with the barrier thickness decrease.

#### 3.1.3. Metal/CaO/BaTiO_3_/Metal Composite Barrier FTJ

The third system studied herein is also a composite FTJ, with a CaO dielectric barrier added to the BaTiO_3_ ferroelectric barrier. The electrodes are of the same generic metal, Me. In this case the asymmetry is ensured just by the presence of the dielectric. We have set the physical parameters to [32]: *λ*_1_ = *λ*_2_ = 1.0 Ǻ, *ε*_1_ = *ε*_2_ = 1, *ε_FE_* = 90, *ε_DE_* = 10, *E_FL_* = *E_FR_* = 3 eV, *φ*_1_ = 5.5 eV, *φ*_2_ = 3.6 eV, *φ_C_* = 1.9 eV, *P* = 40 μC/cm^2^. The effective masses are equal to m_0_ for all materials. We have also kept the thickness of BaTiO_3_ to 2.4 nm and we have varied the thickness of CaO from 0.5 to 3 nm. In comparison to the previous system, in this particular case the barrier is much higher on the dielectric. The calculated conductance and TER ratio are presented in Figure 8.

In contrast to Pt/SrTiO_3_/BaTiO_3_/SrRuO_3_, the conductance and the TER ratio of Me/CaO/BaTiO_3_/Me composite FTJ exhibit an exponential dependence on the dielectric thickness at both 0 K and 300 K. Therefore, qualitatively at least, a 0 K analysis remains valid also at room temperature. In Figure 9, we show the data for a quantitative explanation of conductance and TER ratio behavior with temperature. One can see that the resonance states have a minor contribution to the current; the vast majority of carriers are transported through states around Fermi energy, although there are many resonances below the top of the barrier. Due to the higher dielectric barrier, these resonant states are just weakly coupled to the contacts, hence they show low transmission probability coefficients and they have a modest contribution to conductance.

### 3.2. The Tunneling Wavefunctions. The Wavefunctions of Resonances

In the previous section, it was discussed that the energy-dependent transmission in FTJs has resonant features besides an exponential background. In this section we show that the wavefunctions also exhibit general features, some of which–e.g., those corresponding to resonances–are also observed in other nanostructures like quantum wells, multi-barrier structures, etc. Usually, as a scattering problem, the electron transport in FTJs has two solutions at a given energy: one solution for an incident electron wave coming from the left and the other for the electron wave coming from the right. In the following we will illustrate the wavefunctions at some representative energy values for two FTJs: Pt/BaTiO_3_/SrRuO_3_ and Pt/SrTiO_3_/BaTiO_3_/SrRuO_3_.

#### 3.2.1. The Wavefunctions of Pt/BaTiO_3_/SrRuO_3_ FTJ

We illustrate the wave functions at the Fermi energy (3 eV) and at some resonant energies that can be taken from Figure 10.

It is obvious that up to about 3.5 eV, the transmission probability has an exponential dependence with respect to energy. In Figure 11, we plotted the wavefunctions of a 1.6 nm thick BaTiO_3_ barrier at Fermi energy, as well as the first few resonance energies for both directions of polarizations. In order to compare these wavefunctions we plotted the “normalized” eigenvectors of *AL*, *AR*, and *AD*. In other words, for instance, ${|\Psi_{L,R}\rangle }_{D}$ are divided by $\sqrt{\lambda_{L,R}/2\pi }$. The device region is comprised of the barrier and several layers of contact regions, such that the region outside the device should exhibit a flat electrostatic potential. In practice this is achieved when a few nanometers of contacts are added to the device region. At Fermi energy, the wavefunctions are almost real and their overlap is almost zero. They decay exponentially in the barrier, hence the Simmons or Brinkman formula applies well for states around Fermi energy.

Since the barrier is thin, the resonances are well separated. From Figure 11a,b,d,e we see that the “normalized” ${|\Psi_{L}\rangle }_{D}$ is almost identical with the complex conjugate of “normalized” ${|\Psi_{R}\rangle }_{D}$, however the values of $\lambda_{L,R}/2\pi$ provide the levels of electron density inside the device for the corresponding wavefunction. These states exhibit a confining character within the barrier, in contrast to the states shown at Fermi energy. These solutions belong to resonance and anti-resonance states in the complex wavevector plane [23,25,26]. Moreover, the real and the imaginary parts of ${|\Psi_{L,R}\rangle }_{D}$ can be found in the normalized eigenvectors of *A_D_* (Figure 11c,f). Finally, by analyzing Figure 5 and Figure 11, we notice that just the first resonance would participate to the electron transport at room temperature.

#### 3.2.2. The Wavefunctions of Pt/SrTiO_3_/BaTiO_3_/SrTiO_3_ FTJ

The plots of the wave functions for this composite barrier FTJ are shown in Figure 12. The transmission probability for polarizations is shown in Figure 12a, from which we can extract the resonances that play a role in the electron transport at room temperature. The physical parameters are those that have already been used in the calculations. The effective thickness of the barrier is larger than in simple FTJ presented in the previous subsection; hence the resonances are closely spaced. Still, at Fermi energy the wavefunctions decay exponentially in the barrier and are almost real, while their overlap is almost zero. The wavefunctions of the first four resonances in transmission have a much smaller imaginary part, which is not shown here. As in the previous section, these wavefunctions manifest confining character in the barrier. Moreover, the dielectric induces a much smaller coupling to one of the two contacts, such that the first two resonances in transmission contain a significant background contribution as a decaying wavefunction in the barrier seen in the solution from the left (Figure 12b, positive polarization) and in the solution from the right (Figure 12d, negative polarization). Analyzing Figure 12c,e, one can notice that the overlap between the solution from the left and that from the right is almost zero for the first two resonances in transmission. The next two resonances in transmission, however, can be distinguished from the background, the normalized eigenvectors of *A_L_* and *A_R_* being quite similar, yet the corresponding wavefunctions ${|\Psi_{L}\rangle }_{D}$ and ${|\Psi_{R}\rangle }_{D}$ having quite different amplitudes. The difference in amplitudes starts to close in as we move to higher energies, since at sufficiently higher energy the asymmetry of the barrier will not play any major role in the electron transmission. As a final comment, all four resonances shown here participate to temperature-activated transport (Figure 7). However, the main contribution to temperature-dependent transport is given by the first three resonances, even though the couplings to the contacts of the first two resonances are not as strong as the couplings of the third one, which is further apart in energy.

## 4. Conclusions

In conclusion, the semi-empirical model of electrostatic and NEGF calculations can provide a detailed picture of electron transport in systems with FTJs. It treats on equal footing several transport mechanisms that are usually invoked when studying these devices, such as direct tunneling, thermionic emission, and Fowler-Nordheim tunneling. This feature of NEGF allows us to assess in detailed form the role of temperature in electron transport, and how temperature affects the TER ratio. We have found that for simple or composite barrier BaTiO_3_-based FTJs, the transport through temperature activated resonant (Gamow-Siegert) states may become dominant, affecting both the conductance and the TER ratio; thus, the more resonances are activated and the stronger their coupling to the contacts, the more powerful the effect of temperature. The effect of temperature is obvious in thicker FTJs, since the resonant states–especially those above the barrier–are closer one another, and hence more of them may participate in the transport. More insight into this phenomenology may be acquired by calculating the wavefunctions at any given energy. We show that this is possible for NEGF calculations of single-band transport. Thus, the transport by direct tunneling takes place through states whose wavefunctions have a decaying shape in the barrier. These states belong to the background generated by all resonant states. Furthermore, where the resonances are strong and well-separated, the wavefunctions show confinement character in the barrier. In the intermediate regime, where the resonances are weak, the confining character of the resonance competes with the decaying character of the background. This analysis is valid not only in the linear regime, but also for larger bias voltages characteristic to non-linear regime and relevant to applications; the only change is a rigid shift in the screening charge and in the potential energy of the right contact, as can be seen from Equations (7) and (8). Lastly, we suggest that these results may be usefull in the optimization process of the FTJ design for various applications, particularly at high temperature.

## Figures and Tables

**Figure 1 nanomaterials-12-01682-f001:**
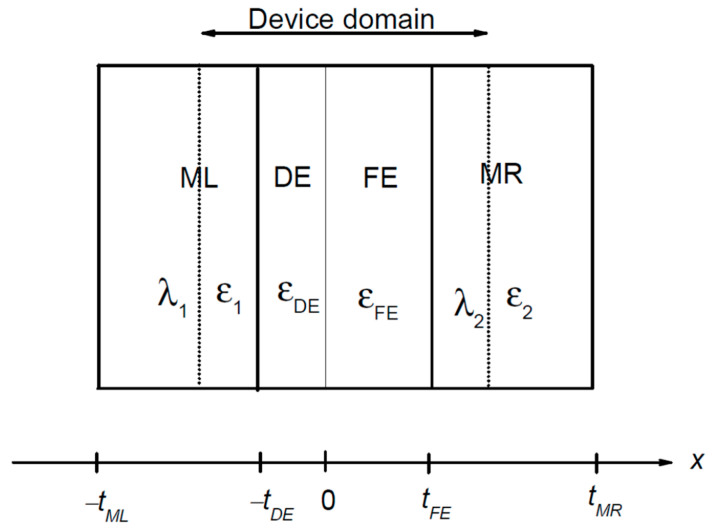
Schematic representation of a FTJ with a composite barrier, consisting of a ferroelectric (FE) layer and a dielectric (DE) layer between two metallic electrodes (ML and MR). The dotted line delimitates the device that is considered in calculations, *λ*_1,2_ are the screening lengths in electrodes, ε_FE,DE_ the relative dielectric constants of the ferroelectric and dielectric layers, ε_1,2_ the relative dielectric constants of the electrodes, and t_FE_,_DE_,_ML_,_MR_ mark the margins of the layers. In the case of a simple FE barrier, the DE layer is not present.

**Figure 2 nanomaterials-12-01682-f002:**
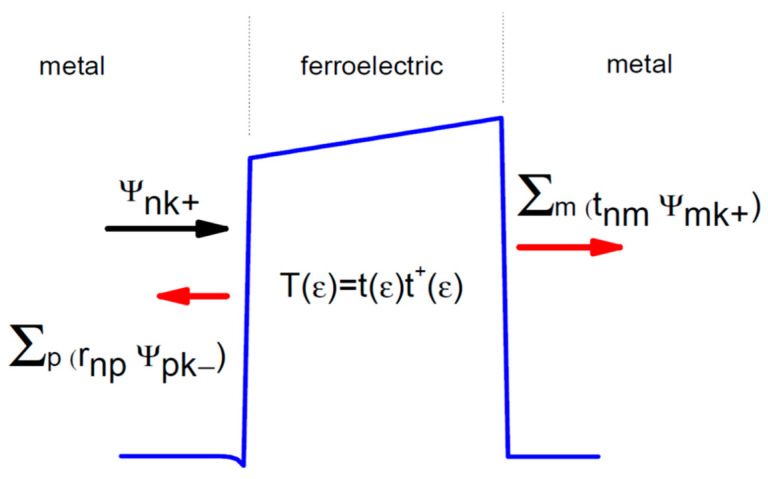
The behavior of an incident Bloch wave Ψ_nk+_ coming from the left. One part is reflected with the coefficient r and the other part is transmitted with a coefficient t. In case of a single-band transport, the coefficients *r* and *t* are simple scalars.

**Figure 3 nanomaterials-12-01682-f003:**
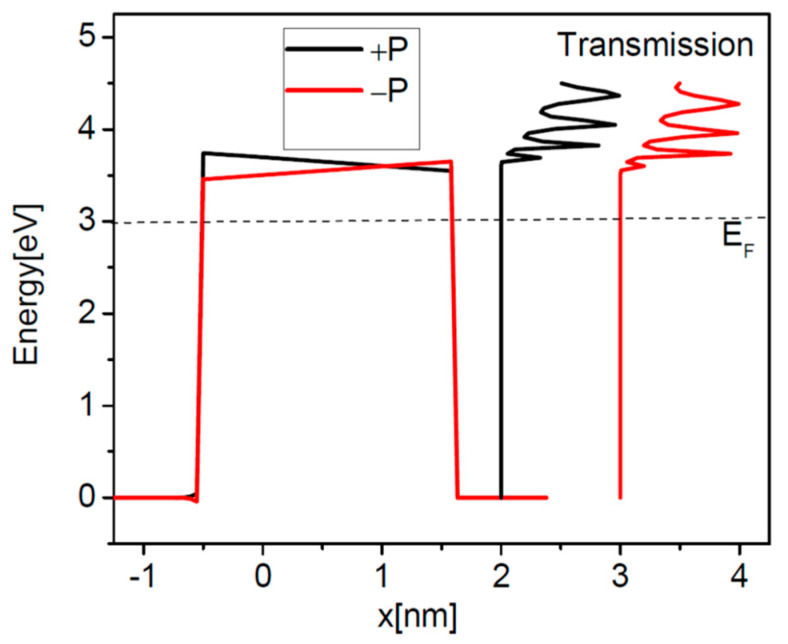
Potential profiles of a BaTiO_3_ barrier of 2 nm thickness between Pt and SrRuO_3_ contacts, for both directions of polarization. Schematically, on the right side are shown the transmission probability coefficients.

**Figure 4 nanomaterials-12-01682-f004:**
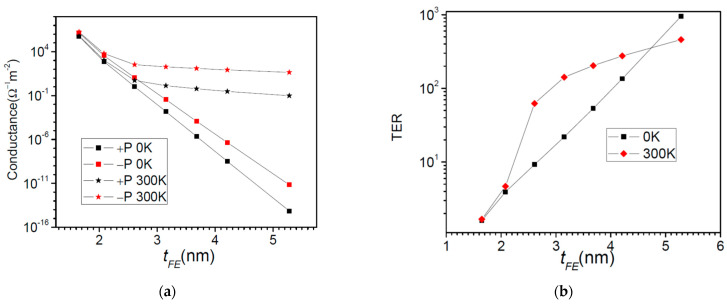
Pt/BaTiO_3_/SrRuO_3_ FTJ: (**a**) Conductance, and (**b**) TER ratio as a function of the barrier thickness at 0 K and 300 K.

**Figure 5 nanomaterials-12-01682-f005:**
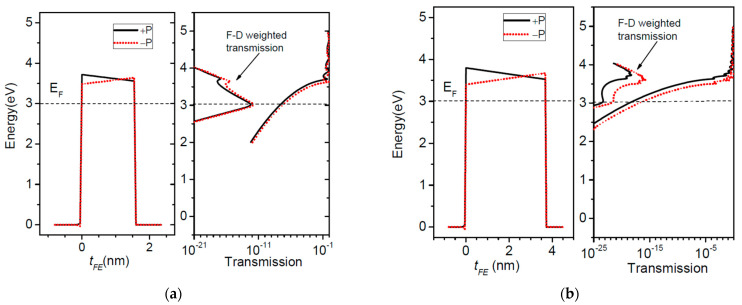
Pt/BaTiO_3_/SrRuO_3_ FTJ: the barrier profile, the transmission probability coefficient, and the Fermi-Dirac weighted transmission at 300 K for two barrier thicknesses—(**a**) 1.6 nm and (**b**) 3.5 nm. The dashed line in the graphs indicates the Fermi energy.

**Figure 6 nanomaterials-12-01682-f006:**
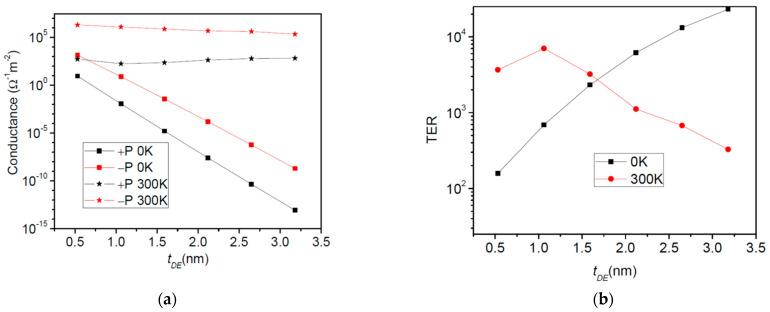
Pt/SrTiO_3_/BaTiO_3_/SrRuO_3_ composite FTJ: (**a**) Conductance and (**b**) TER ratio as a function of the dielectric thickness at 0 K and 300 K.

**Figure 7 nanomaterials-12-01682-f007:**
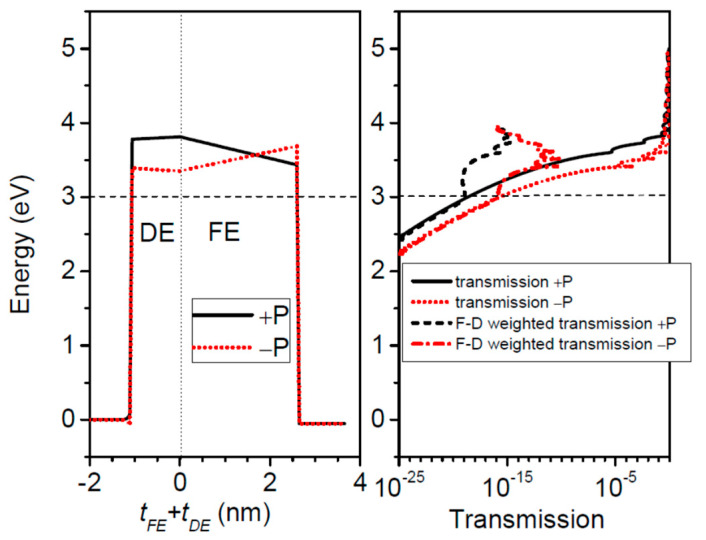
Pt/SrTiO_3_/BaTiO_3_/SrRuO_3_ composite FTJ: the barrier profile, the transmission probability coefficient, and the Fermi-Dirac weighted transmission at 300 K; t_FE_ = 2.5 nm, t_DE_ = 1 nm. The dashed line in the graphs indicates the Fermi energy.

**Figure 8 nanomaterials-12-01682-f008:**
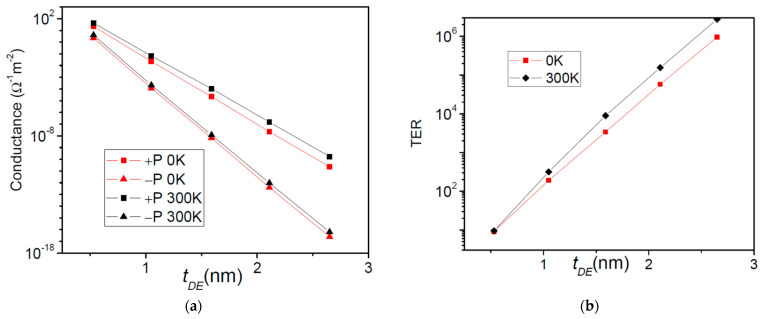
Me/CaO/BaTiO_3_/Me composite FTJ: (**a**) Conductance and (**b**) TER ratio as a function of the dielectric thickness at 0 K and 300 K.

**Figure 9 nanomaterials-12-01682-f009:**
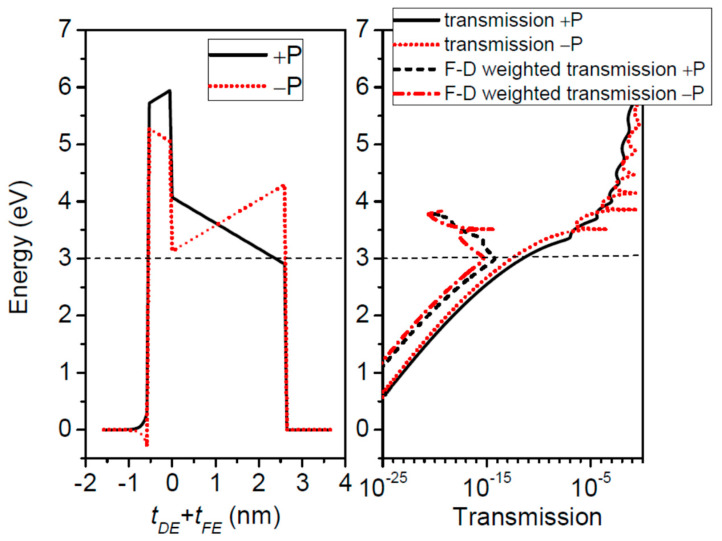
The Me/CaO/BaTiO_3_/Me composite FTJ: the barrier profile, the transmission probability coefficient, and the Fermi-Dirac weighted transmission at 300 K; *t_FE_* = 2.5 nm, *t_DE_* = 0.5 nm. The dashed line in the graphs indicates the Fermi energy.

**Figure 10 nanomaterials-12-01682-f010:**
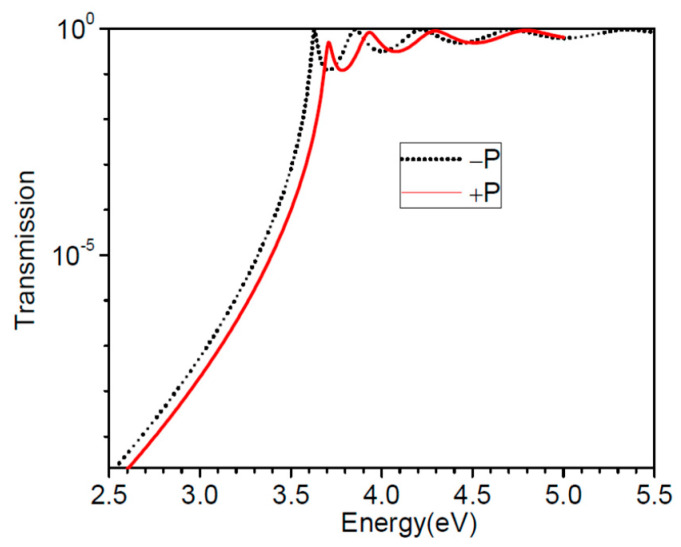
Transmission probability for a BaTiO_3_ barrier of 1.6 nm thickness between Pt and SrRuO_3_ metallic contacts for both polarization directions.

**Figure 11 nanomaterials-12-01682-f011:**
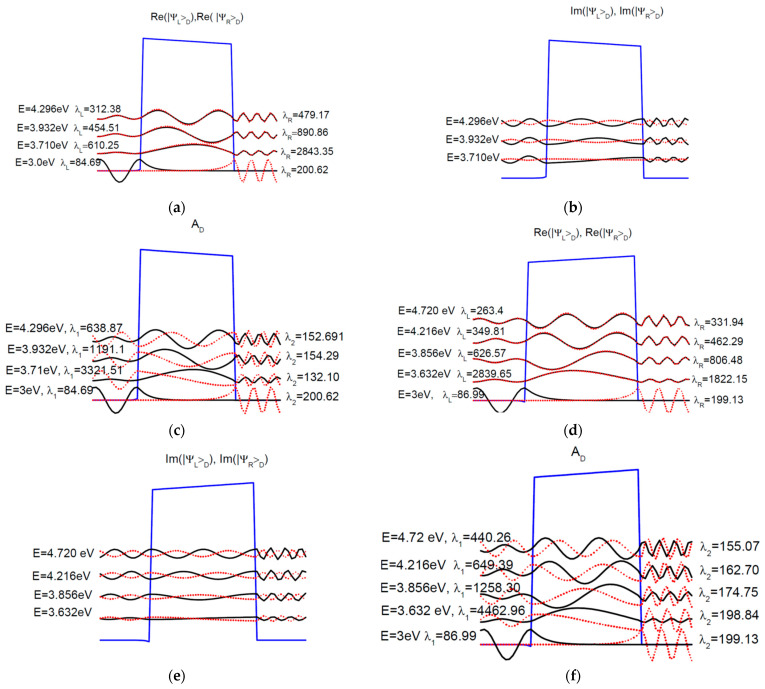
Representative wavefunctions of BaTiO_3_ FTJ as normalized eigenvectors of *A_L_* (solid black line), *A_R_* (dotted red line). They are scaled down by a factor $\sqrt{\lambda_{L,R}/2\pi }$, where *λ_L,R_* is the corresponding eigenvalues of *A_L_*, *A_R_*. In addition we illustrate the normalized eigenvectors of *A_D_* with eigenvalue *λ*_1_ (solid black line) and *λ*_2_ (dotted red line). The normalized eigenvectors of *A_L_*, *A_R_* are as follows: (**a**)—real part, positive ferroelectric polarization; (**b**)—imaginary part, positive ferroelectric polarization; (**d**)—real part, negative ferroelectric polarization; (**e**)—imaginary part, negative, ferroelectric polarization. The normalized eigenvectors of *A_D_* are as follows: (**c**)—positive ferroelectric polarization; (**f**)—negative ferroelectric polarization.

**Figure 12 nanomaterials-12-01682-f012:**
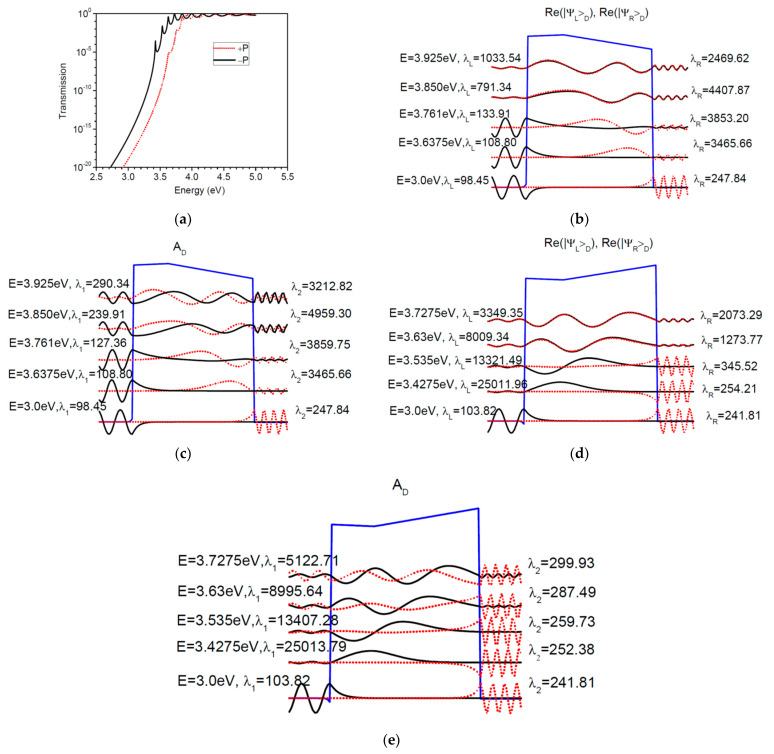
Pt/SrTiO_3_/BaTiO_3_/SrTiO_3_ composite FTJ (1 nm of SrTiO_3_ and 2.4 nm of BaTiO_3_). (**a**) Transmission probability coefficient for positive (dotted red line) and negative (solid black line) ferroelectric polarization; (**b**–**e**) Like in Figure 11 the representative wavefunctions are the normalized eigenvectors of *A_L_* and *A_R_* as well as the normalized eigenvectors of *A_D_*. The real part of normalized eigenvectors of *A_L_* (solid black line) *A_R_* (dotted red line) are as follows: (**b**)—positive ferroelectric polarization; (**d**)—negative ferroelectric polarization. The normalized eigenvectors of *A_D_* with eigenvalue *λ*_1_ (solid black line) and *λ*_2_ (dotted red line) are as follows: (**c**)—positive ferroelectric polarization; (**e**)—negative ferroelectric polarization.

## Data Availability

Not applicable.

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
