# Peer review of "Insights into Electron Transport in a Ferroelectric Tunnel Junction"

_nanomaterials, 2022, doi:10.3390/nano12101682_

Round 1

Reviewer 1 Report

Report on MDPI 1596462: “Insights into Electron Transport in a Ferroelectric Tunnel Junction”  by  Sandu et al. 

The conductance  of  various  layer system configurations which include  a BaTiO3 ferroelectric  barrier for both directions of polarization  P is calculated, with the use of a semiempirical model. The ferroelectric polarization is considered as a mean field order parameter which changes the shape of the barrier in the tunnel system. 

 These tunnel junctions  have been extensively studied in the ’80ties   and  ’90ties and the analysis of electron transport in such systems has reached a level of sophistication which does not appear  as a background for  the present work.  There is also   a flood of literature  on ferromagnetic  (spin dependent tunnel junctions) and ferroelectric  barriers, MOS structures, transport across  contacted quantum dots  and Kondo interaction,  which has produced a consolidated amount of expertise on electron interaction in the barrier with  metallic contacts in open systems.  The tool is the Keldysh formulation and Dyson equation for the Green functions with the seminal  works  by Meir, Wingreen and Lee [ e.s. Meir, Wingreen,  Phys. Rev. Lett. 68, 2512 (1992)].   Section 2 is devoted to a primitive  theoretical approach  which has evolved incredibly in these four decades since then.  I am not an expert of the state of the art now, but there are DFT numerical  turnkey codes available.  There is also ambiguity in the approach, because the Green’s function is  a propagator in an interacting system  in which the full wave function is not attainable and a reduced  effective wavefunction, as the one presented in Section 3.2  in the form of a single electron barrier problem of the early quantum mechanics days,  is a misleading  concept superseeded by the Green function density matrix approach.   

I found also very initial  difficulties in reading, as  there is an electric field in  the l.h.s. of   Eq.3 , while   tau_s appearing on the r.h.s. is defined as a screening charge. Same for   Eq. 4,5 where  the built-in voltage bias  VB is defined but does not appear.  Further on, what was an electric field becomes an energy

(e.g. in Eq. 6). 

 The hamiltonian of Eq.10 ( not well defined) is  not a tight binding approach  but some kind of effective mass 

approximation. I hope that the orders of magnitudes 10^-25 presented in the plots  are not the ones of the numerical calculation, because, otherwise,  they are too small to be reliable. 

As for the results, it is likely that , by increasing the  temperature, the TER decreases, and the justification given ( larger weight due to the scattering states above the P -dependent effective barrier) appears likely.

In conclusion, this work does not appear as being up to  date nor does it  to contribute in understanding experimental findings, as no comparison with experiment is  attempted. 

It should not be published.

Author Response

We wish to thank all Reviewers for their comments and suggestions aiming to improve our work. As a result of their effort we have undertaken a thorough revision of our manuscript. Here below, we highlight the main changes in the manuscript and also provide the necessary explanations to some points raised by the Reviewers.

Dr. Titus Sandu

March 17, 2022.

Reviewer 1

Report on MDPI 1596462: “Insights into Electron Transport in a Ferroelectric Tunnel Junction”  by  Sandu et al. 

The conductance  of  various  layer system configurations which include  a BaTiO3 ferroelectric  barrier for both directions of polarization  P is calculated, with the use of a semiempirical model. The ferroelectric polarization is considered as a mean field order parameter which changes the shape of the barrier in the tunnel system. 

 These tunnel junctions  have been extensively studied in the ’80ties   and  ’90ties and the analysis of electron transport in such systems has reached a level of sophistication which does not appear  as a background for  the present work. 

Authors' response: In our work we develop a simple and intuitive approach in order to highlight the importance of the resonance states appearing under particular conditions in FTJ systems and of finite temperature effect on the conduction properties. Our work follows the recent developments in the field of MFTJs like Ref. 14 [Chang et al., Theoretical Approach to Electroresistance in Ferroelectric Tunnel Junctions, Phys. Rev. Appl. 2017, 7, 024005].

Also our work makes a connection between NEGF and the resonance state expansion for open quantum systems [Refs 23-26].

To our knowledge this is the first approach that emphasizes the role of the resonance states located well above Fermi level, resonance states that may become important for electron conduction at room temperature.

It came to our attention a quite recent work in which the resonance states play a crucial role in FTJ: [Su et al., “Resonant Band Engineering of Ferroelectric Tunnel Junctions”, PRB 104, L060101 (2021)]. However, unlike our case, this paper treats the resonant quasi-bond states that are close to Fermi level, hence they are reachable at low temperature.

-------------------------------------------------------------

There is also   a flood of literature  on ferromagnetic  (spin dependent tunnel junctions) and ferroelectric  barriers, MOS structures, transport across  contacted quantum dots  and Kondo interaction,  which has produced a consolidated amount of expertise on electron interaction in the barrier with  metallic contacts in open systems.  The tool is the Keldysh formulation and Dyson equation for the Green functions with the seminal  works  by Meir, Wingreen and Lee [ e.s. Meir, Wingreen,  Phys. Rev. Lett. 68, 2512 (1992)].   Section 2 is devoted to a primitive  theoretical approach  which has evolved incredibly in these four decades since then.  I am not an expert of the state of the art now, but there are DFT numerical  turnkey codes available.  There is also ambiguity in the approach, because the Green’s function is  a propagator in an interacting system  in which the full wave function is not attainable and a reduced  effective wavefunction, as the one presented in Section 3.2  in the form of a single electron barrier problem of the early quantum mechanics days,  is a misleading  concept superseeded by the Green function density matrix approach.   

 Authors' response: Our work is not about the latest achievements in the NEGF theory. Of course there are a plethora of flavors of NEGF - ab-initio, ab-initio-like, semi-emprical, etc.,-, treating more or less exactly one or more physical aspects of a given problem. The work mentioned above [Phys. Rev. Lett. 68, 2512 (1992)] provides a formula for a device with interacting electrons in the device region. In our manuscript we considered just the tunneling, i.e., Eq. (7) in the work of Meir and Wingreen for non-interacting electrons. This is also discussed in Ref. 22 [Single and multiband modeling of quantum electron transport through layered semiconductor devices, J. Appl. Phys. 1997, 81, 7845], where the authors have shown that the tunneling transport is basically the Lee and Fisher formula obtained in a scattering approach [D. S. Fisher and P. A. Lee, Phys. Rev. B 23, 6851, (1981)].

We think that the referee did not fully understand our results about the retrieving of the wave function from the Green’s function. As it is stated also in Ref. 28 [Transmission eigenchannels from nonequilibrium Green’s functions, Phys. Rev. B 2007, 76, 115117], in principles it is not possible to have the scattering wave function from the spectral function. This can be seen from the recent work on the ab-intio transport code TRANSIESTA [Eq. 16 in ¨Improvements on non-equilibrium and transport Green function techniques: The next-generation transiesta¨, Computer Physics Communications 212 (2017), p. 8–24], where one deals with block matrices in general case. Nonetheless, just for one band models it happens that the scattering wave function is retrieved from the spectral function.

--------------------------------------------------

I found also very initial  difficulties in reading, as  there is an electric field in  the l.h.s. of   Eq.3 , while   tau_s appearing on the r.h.s. is defined as a screening charge. Same for   Eq. 4,5 where  the built-in voltage bias  VB is defined but does not appear.  Further on, what was an electric field becomes an energy

(e.g. in Eq. 6). 

Authors' response: We corrected all these inconsistencies and omissions.

 The hamiltonian of Eq.10 ( not well defined) is  not a tight binding approach  but some kind of effective mass approximation. I hope that the orders of magnitudes 10^-25 presented in the plots  are not the ones of the numerical calculation, because, otherwise,  they are too small to be reliable. 

 Authors' response: In the appendix we have shown the recipe of converting a single-band effective mass Hamiltonian into a tight-binding Hamiltonian. It follows closely Ref. 22 [ J. Appl. Phys. 1997, 81, 7845]

In general, in the tight binding reprezentation, the Hamiltonian has a block matrix form, more precisely, a block tridiagonal form as in eq. (10). For a single band model the block tridiagonal matrix form of the H becomes a simple tridiagonal matrix. We have explained the terms in Eq.(10) and corrected the text.

The plots down to 10^-25 were used to illustrate the effect of temperature in all studied cases. We carefully checked the validity of our numerical results. First, one sign of a numerical error would be the numerical noise, which is not present. Secondly, numbers of 10^-25 magnitude are obtained by mere multiplication in which one does not lose numerical precision (Eqs. 17-20). Thirdly, these small numbers do not count into the final results regarding the conductance since an integration over transverse wave-vector and energy are performed.

New text added at line 164 (revised manuscript):

"In general, in the tight binding reprezentation, the Hamiltonian has a block matrix form, more precisely, a block tridiagonal form as in eq. (10). For a single band model the block tridiagonal matrix form of the H becomes a simple tridiagonal matrix."

-------------------------------------------------------------

As for the results, it is likely that , by increasing the  temperature, the TER decreases, and the justification given ( larger weight due to the scattering states above the P -dependent effective barrier) appears likely.

Authors' response: We have not argued about TER decrease by increasing the temperature. We rather emphasized the non-monotonic behavior of TER with respect to barrier thickness, which is a temperature effect.

In conclusion, this work does not appear as being up to  date nor does it  to contribute in understanding experimental findings, as no comparison with experiment is  attempted. 

 It should not be published.

Submission Date

28 January 2022

Date of this review

12 Feb 2022 13:07:41

Reviewer 2 Report

The authors apply the non-equilibrium Green function and incorporate quantum resonance effects to assess if existing conduction models accurately predict FTJ operational performance (such as conductance and tunneling electroresistance ratio), particularly as it relates to temperature and barrier thickness. Overall the paper goes into some depth in its derivation of the approach and provides nice insight for theoreticians and physicists on the method of calculation. Its impact may be somewhat limited to more theoretically inclined groups, however, because the present state of the manuscript does not compare the simulation results to experimental data, nor does it reach a closed-form solution that would be invaluable to device designers and engineers.

If the authors can try to establish a clearer case on the relevancy of these results to device operation, as outlined by several suggestions below, I think the manuscript would improve substantially and widen its impact and audience. The reviewer does not expect all points to be addressed with scores of new simulations, as this would likely become too exhaustive in the reviewer’s opinion, but hopes the authors can make a good faith attempt to incorporate a few of these suggestions that can further broaden the manuscript’s appeal and impact to a wider audience before publication:

  1. The authors state on page 12 that the calculation of the conductance was performed with an applied bias voltage of 1 μV for the 300 K results in Figure 4a. Was this bias applied to all calculations? The bias should be clearly stated for each simulation. A bias voltage of 1 μV, even for a read condition in an FTJ, is extremely low. Since the authors are arguing that their model incorporating quantum resonance effects yields critical information related to FTJ operating characteristics, shouldn’t the simulation conditions be chosen to assess the relative accuracy of the conduction mechanisms (direct tunneling, Fowler Nordheim, thermionic emission) at more relevant applied biases (100’s mV)?
  2. The work would benefit if the authors directly compared their simulation results to existing experimental data to further support their arguments that incorporating resonance effects can explain shortcomings in existing FTJ operational behavior. If existing experimental data can be well-modeled by the existing conduction mechanisms, the authors need to more concretely explain why the incorporation of resonance effects (i.e. under what conditions) are of importance to the development of the technology and application of FTJs.
  3. Ferroelectric-metal and ferroelectric-dielectric interfaces often contain a high number of defect states. How might interfacial and bulk trap states influence the relative importance of these resonance effects?
  4. Is it possible for the authors to derive an analytical approximation for the J(V) equation that incorporates the resonant features or is the model only possible by numerical simulation? If the authors could encapsulate these effects in closed form and as a function of bias, this would greatly facilitate the ability of device engineers to apply their model to experimental data. In general, it would be fascinating if the authors could, by reverse-engineering their simulation results, yield closed-form relationships that can be used to predict performance without the need for high-computational power.
  5. Since the incorporated resonance effects are conjectured to be helpful to explain the temperature-dependent characteristics of FTJ’s, the inclusion of only two temperatures (0K and 300K) really limits what can be predicted from this model. Is there anyway the authors can make a temperature plot with all other relevant parameters constant?

Author Response

We wish to thank all Reviewers for their comments and suggestions aiming to improve our work. As a result of their effort we have undertaken a thorough revision of our manuscript. Here below, we highlight the main changes in the manuscript and also provide the necessary explanations to some points raised by the Reviewers.

Dr. Titus Sandu

March 17, 2022.

Reviewer 2

The authors apply the non-equilibrium Green function and incorporate quantum resonance effects to assess if existing conduction models accurately predict FTJ operational performance (such as conductance and tunneling electroresistance ratio), particularly as it relates to temperature and barrier thickness. Overall the paper goes into some depth in its derivation of the approach and provides nice insight for theoreticians and physicists on the method of calculation. Its impact may be somewhat limited to more theoretically inclined groups, however, because the present state of the manuscript does not compare the simulation results to experimental data, nor does it reach a closed-form solution that would be invaluable to device designers and engineers.

If the authors can try to establish a clearer case on the relevancy of these results to device operation, as outlined by several suggestions below, I think the manuscript would improve substantially and widen its impact and audience. The reviewer does not expect all points to be addressed with scores of new simulations, as this would likely become too exhaustive in the reviewer’s opinion, but hopes the authors can make a good faith attempt to incorporate a few of these suggestions that can further broaden the manuscript’s appeal and impact to a wider audience before publication:

  1. The authors state on page 12 that the calculation of the conductance was performed with an applied bias voltage of 1 μV for the 300 K results in Figure 4a. Was this bias applied to all calculations? The bias should be clearly stated for each simulation. A bias voltage of 1 μV, even for a read condition in an FTJ, is extremely low. Since the authors are arguing that their model incorporating quantum resonance effects yields critical information related to FTJ operating characteristics, shouldn’t the simulation conditions be chosen to assess the relative accuracy of the conduction mechanisms (direct tunneling, Fowler Nordheim, thermionic emission) at more relevant applied biases (100’s mV)?

Authors' response: The applied bias voltage is of 100 μV (10*-4 V in the text). In order to compare the conductivity at finite temperature (300K) with the results at T=0K, we have chosen a bias voltage of 100 μV to ensure the linearity of the regime at finite temperature. The calculation of conductivity at T=0K was performed with Eq. (20), while at 300 K the conductivity was calculated with Eq. (19). At finite bias voltages (100’s mV) the barrier has to be shifted accordingly up or down. We have not assessed the relative accuracy of the conduction mechanisms (direct tunneling, Fowler-Nordheim, thermionic emission) since our goal was to show the role of the resonance states on TER ratio. In doing so, it was found that the TER ratio has a non-monotonic dependence on thickness of ferroelectric.

The relative accuracy of these conduction mechanisms (direct tunneling, Fowler Nordheim, thermionic emission) can be performed by comparing the results of the analytical formulas for those mechanisms with the results obtained by using Eq. (19) with energy integration over relevant intervals: for direct tunneling from 0 eV to the beginning of triangular profile, for Fowler-Nordheim from the beginning of triangular profile to the top of it, and for thermionic emission from the top of the triangular profile to infinity.

  1. The work would benefit if the authors directly compared their simulation results to existing experimental data to further support their arguments that incorporating resonance effects can explain shortcomings in existing FTJ operational behavior. If existing experimental data can be well-modeled by the existing conduction mechanisms, the authors need to more concretely explain why the incorporation of resonance effects (i.e. under what conditions) are of importance to the development of the technology and application of FTJs.

Authors' response: A non-monotonic behavior of TER ratio with respect to ferroelectric thickness was observed experimentally in Ref. 31 [Wang et al., “Overcoming the Fundamental Barrier Thickness limits of Ferroelectric Tunnel Junctions through BaTiO_3 /SrTiO_3 Composite Barriers”, Nano Lett. 2016, 16, 3911-3918] for both simple and composite FTJ. We have mentioned it in our manuscript. Wang et al., performed their analysis with the Brinkman model, which is 0 K model, and attributed the non-monotonic behavior of TER ratio to the play of large leakage currents for thin barriers with the noisy current measurements for thick barriers. We think that resonance states have a contribution to the non-monotonic behavior of TER ratio. Regarding the resonance states, it came to our attention the recent paper, Su et al., “Resonant Band Engineering of Ferroelectric Tunnel Junctions”, PRB 104, L060101 (2021), in which the resonant states enhance the TER ratio. However, unlike our case, in this recent paper there are resonant quasi-bond states that are close to Fermi level, hence they are reachable at low temperatures.

  1. Ferroelectric-metal and ferroelectric-dielectric interfaces often contain a high number of defect states. How might interfacial and bulk trap states influence the relative importance of these resonance effects?

Authors' response: Certainly, this aspect, i.e. defects at interfaces, raised by the Reviewer, is of utmost importance for the FTJs performance, considering defects unavoidable presence in a practical device, as well as their strong effect on band offsets and all transport properties. However, the development of an analytic model to include defects and associated strains, besides the finite temperature effect deserves a detailed separate study. The resonance state expansion pertains the development of a transfer Hamiltonian model in which the inclusion of scattering from defects and from phonons is made by self-energies like the treatments of semiconductor heterostructures, see for instance Zou and Chao, “Inelastic electron resonant tunneling through a double-barrier nanostructure”, Phys. Rev. Lett, 69, 3224,(1992) for electron-phonon scattering and Fertig et al., Elastic-scattering effects on resonant tunneling in double-barrier quantum-well structures, Phys. Rev., B41, 3596, (1990) for elastic scattering on defects.

  1. Is it possible for the authors to derive an analytical approximation for the J(V) equation that incorporates the resonant features or is the model only possible by numerical simulation? If the authors could encapsulate these effects in closed form and as a function of bias, this would greatly facilitate the ability of device engineers to apply their model to experimental data. In general, it would be fascinating if the authors could, by reverse-engineering their simulation results, yield closed-form relationships that can be used to predict performance without the need for high-computational power.

Authors' response: We agree with the Reviewer's suggestion, but this could be the subject of another paper. Analytic expressions for resonant states are available for rectangular barrier [Yamada et al., Quantum-shutter approach to tunneling time scale with wave-packets, Phys. Rev. A, 72, 012106,

(2005)]. Hence, besides the inclusion of scattering processes, as it was discussed the previous comment, a resonant states approach would facilitate also a straightforward time-dependent evolution of the system.

There is also a resonance state expansion for trapezoidal barriers, but their expressions are cumbersome [D. Juhasz et al., “Convergence and completeness for square-well Stark resonant state expansion”, J. Math. Phys., 59, 113501, (2018)].

  1. Since the incorporated resonance effects are conjectured to be helpful to explain the temperature-dependent characteristics of FTJ’s, the inclusion of only two temperatures (0K and 300K) really limits what can be predicted from this model. Is there anyway the authors can make a temperature plot with all other relevant parameters constant?

Authors' response: The calculations were performed at two reference temperatures (0K and 300K) because many theoretical evaluations of FTJs are performed at 0K and most of the experimental characterizations are made at room temperature. The choice to show just these two specific temperatures was made in order to stress the role of temperature as a critical parameter for device characterization. We may generate a temperature plot, but in order to have reliable results it may be quite costly in time.

Submission Date

28 January 2022

Date of this review

04 Mar 2022 16:07:38

Reviewer 3 Report

The paper reports results of a theoretical analysis of the tunneling electroresistance (TER) effects on ferroelectric tunnel junctions (FTJs). The authors found that the conductance and the TER ratio of FTJs is affected through temperature activated resonance states. The results are useful and informative to optimize the FTJs design for various applications. However, the manuscript has not been carefully written. I recommend that the authors carefully check the texts, equations, and figures, and the following points to be addressed.

The TER ratios defined by Eq. (1) would be inconsistent with those shown in Figs. 4b, 6b, and 8b because they are smaller than 1 in the equation, whereas they are larger than 1 in the figures. The denominator of Eq. (1) seems to be not JON but JOFF.

dFE” appeared in lines 119 and 120 is not defined in texts and seems to be typos of “tDE”.

VBI” in Eq. (5) seems to be a typo of “VB”.

“V” in line 134 should be italic.

“V” and “B” of “VB” in line 134 should be italic and subscript, respectively.

When eliminating EFE and EDE from Eq. (4) and (5), the sign of τS in Eq. (7) becomes minus.

The conductance G defined by Eq. (20) should be expressed using another symbol because “G” is confusable with the Green functions. The title of the vertical axis “G” of Fig. 8a should be changed to “Conductance” as same with Figs. 4a and 6a.

“Pt/SrTiO3/BaTiO3/SrTiO3” in lines 446 and 474 seems to be typos of “Pt/SrTiO3/BaTiO3/SrRuO3”.

Author Response

We wish to thank all Reviewers for their comments and suggestions aiming to improve our work. As a result of their effort we have undertaken a thorough revision of our manuscript. Here below, we highlight the main changes in the manuscript and also provide the necessary explanations to some points raised by the Reviewers.

Dr. Titus Sandu

March 17, 2022.

Reviewer 3

The paper reports results of a theoretical analysis of the tunneling electroresistance (TER) effects on ferroelectric tunnel junctions (FTJs). The authors found that the conductance and the TER ratio of FTJs is affected through temperature activated resonance states. The results are useful and informative to optimize the FTJs design for various applications. However, the manuscript has not been carefully written. I recommend that the authors carefully check the texts, equations, and figures, and the following points to be addressed.

The TER ratios defined by Eq. (1) would be inconsistent with those shown in Figs. 4b, 6b, and 8b because they are smaller than 1 in the equation, whereas they are larger than 1 in the figures. The denominator of Eq. (1) seems to be not JON but JOFF.

Authors' response: We replaced JON by JOFF

dFE” appeared in lines 119 and 120 is not defined in texts and seems to be typos of “tDE”.

Authors' response:

New text (lines 122-124 revised manuscript): "We assume that the dielectric and ferroelectric are located at x between -tDE and 0 and x between 0 and tFE, respectively, while the metallic contact ML is at x <- tDE and the metallic contact MR is at x>tFE, as illustrated in Figure 1."

replaces

Old text: " If the thickness of the dielectric is fixed to tDE and that of the ferroelectric to tFE, we assume that the dielectric and ferroelectric are located for x between -dFE and 0 and for x between 0 and tFE, respectively, while the metallic contact ML is at x <- dFE and the metallic contact MR is at x>tFE, as illustrated in Figure 1.
---------------------------------------

VBI” in Eq. (5) seems to be a typo of “VB”.

“V” in line 134 should be italic.

“V” and “B” of “VB” in line 134 should be italic and subscript, respectively.

When eliminating EFE and EDE from Eq. (4) and (5), the sign of τS in Eq. (7) becomes minus.

Authors' response: We corrected these errors.

The conductance G defined by Eq. (20) should be expressed using another symbol because “G” is confusable with the Green functions. The title of the vertical axis “G” of Fig. 8a should be changed to “Conductance” as same with Figs. 4a and 6a.

“Pt/SrTiO3/BaTiO3/SrTiO3” in lines 446 and 474 seems to be typos of “Pt/SrTiO3/BaTiO3/SrRuO3”.

  Authors' response: We corrected also these errors.

Submission Date

28 January 2022

Date of this review

05 Mar 2022 23:39:10

Reviewer 4 Report

I have read the manuscript of the paper entitled “Insights into Electron Transport in a Ferroelectric Tunnel Junction” written by Titus Sandu, Catalin Tibeica, Rodica Plugaru, Oana Nedelcu and Neculai Plugaru. 

In this paper, a theoretical work on the analysis of the operating efficiency of Ferroelectric Tunnel Junction (FTJ) devices, which are expected to be used as a promising nonvolatile memory device of the future, is discussed.

The authors calculate the physical properties of FTJs using a semi-empirical method that can model the device structure with only a few effective parameters.

In the previous studies on this device, three possible processes have been considered for the conductive charge to pass through the ferroelectric layer.  Two of them are quantum mechanical processes in which the charge passes through the ferroelectric barrier layer by a tunneling process, and the other is a thermal excitation process in which the charge is thermally excited via a Fermi-Dirac distribution and overcomes the barrier potential.

The authors raised a concern that this thermal excitation model assumes that the transmission probability of a charge overcoming a barrier potential is universally unity regardless of the excess energy.  In the present study, they applied semi-empirical quantum mechanical calculations to the high-temperature electrons overcoming such a barrier and found that the transmission probability exhibits a characteristic resonant energy structure related to the thickness of the insulating layer.

Since such quantum effects on thermally excited electrons have not been discussed in previous studies, the importance of this paper in discussing the validity of such a mechanism is highly appreciated.  However, there are still many parts of the manuscript that are difficult to read or insufficiently discussed.  Below I have listed the problems I noticed, and I hope you will find them helpful in making the paper more understandable.

1) Chapter number 2.1 is duplicated.

2) The beginning of the paragraph does not seem to match the logical structure.  In particular, the line immediately following the formulas.

3) The figures are too small.  The underlined and dotted lines in Figures 5, 7, and 9, and the scale of the horizontal axis are hardly grasped.

4) Information in Figure 1 is not sufficient. The parameters used in the calculation, x, λ, etc., need to be described.

5) The paper deals with four-layer devices (metal/dielectric/ferroelectric/metal) in addition to three-layer devices (metal/dielectric/metal), but the four-layer devices are not considered in Fig. 1.  Nontheless, the description of the four-layer structure starts at the beginning of the Theoretical Background section.  The way of description will confuse the readers.

6) p 134, VB?

7) Strange matrix element in Eq(10)

8) The coherence of the electron wave function is important for the appearance of resonant structures reflecting the layered geometry. However, the temperature effect is only incorporated into the distribution function of eq(18), and the dephasing of the wave function does not seem to be taken into account.  According to Fig. 5 and Fig. 11, even the lowest energy resonance line is located about 0.7 eV above the Fermi surface (3.71 eV - 3.0 eV).  In terms of temperature, this is an extremely high value exceeding 8000K. No matter how thin the barrier layer is, the reader may wonder how electrons with such high temperature can maintain sufficient coherence.  I think a reasonable explanation is necessary.

Author Response

We wish to thank all Reviewers for their comments and suggestions aiming to improve our work. As a result of their effort we have undertaken a thorough revision of our manuscript. Here below, we highlight the main changes in the manuscript and also provide the necessary explanations to some points raised by the Reviewers.

Dr. Titus Sandu

March 17, 2022.

Reviewer 4

I have read the manuscript of the paper entitled “Insights into Electron Transport in a Ferroelectric Tunnel Junction” written by Titus Sandu, Catalin Tibeica, Rodica Plugaru, Oana Nedelcu and Neculai Plugaru. 

In this paper, a theoretical work on the analysis of the operating efficiency of Ferroelectric Tunnel Junction (FTJ) devices, which are expected to be used as a promising nonvolatile memory device of the future, is discussed.

The authors calculate the physical properties of FTJs using a semi-empirical method that can model the device structure with only a few effective parameters.

In the previous studies on this device, three possible processes have been considered for the conductive charge to pass through the ferroelectric layer.  Two of them are quantum mechanical processes in which the charge passes through the ferroelectric barrier layer by a tunneling process, and the other is a thermal excitation process in which the charge is thermally excited via a Fermi-Dirac distribution and overcomes the barrier potential.

The authors raised a concern that this thermal excitation model assumes that the transmission probability of a charge overcoming a barrier potential is universally unity regardless of the excess energy.  In the present study, they applied semi-empirical quantum mechanical calculations to the high-temperature electrons overcoming such a barrier and found that the transmission probability exhibits a characteristic resonant energy structure related to the thickness of the insulating layer.

Since such quantum effects on thermally excited electrons have not been discussed in previous studies, the importance of this paper in discussing the validity of such a mechanism is highly appreciated.  However, there are still many parts of the manuscript that are difficult to read or insufficiently discussed.  Below I have listed the problems I noticed, and I hope you will find them helpful in making the paper more understandable.

1) Chapter number 2.1 is duplicated.

Authors' response: We have corrected the Chapter number: 2.2 Transport by NEGF

2) The beginning of the paragraph does not seem to match the logical structure.  In particular, the line immediately following the formulas.

Authors' response: Subsection "2.1 The profile of potential barrier" and Subsection "2.2 Transport by NEGF" follow the structure presented in the last paragraph in Introduction. We are not able to detect the place refered in comment 2).

3) The figures are too small.  The underlined and dotted lines in Figures 5, 7, and 9, and the scale of the horizontal axis are hardly grasped.

Authors' response: The figures have been enlarged and corrected according to the Referee's comments.

4) Information in Figure 1 is not sufficient. The parameters used in the calculation, x, λ, etc., need to be described.

Authors' response: The parameters used in the calculations have been marked on the figure and explained in the caption.

Figure 1. Schematic representation of a FTJ with composite barrier, consisting of a ferroelectric (FE) layer and a dielectric (DE) layer between two metallic electrodes (ML and MR). The dotted lines delimitate the electronic device that is considered in calculations. lambda_1,2 is the screening length, epsilon_FE/DE the relative dielectric constants of the ferroelectric/dielectric layer, epsilon1,2 the relative dielectric constants of the electrodes, and t_FE,DE,ML,MR mark the margins of the layers. In the case of a simple FE barrier, the DE layer is not present.

5) The paper deals with four-layer devices (metal/dielectric/ferroelectric/metal) in addition to three-layer devices (metal/dielectric/metal), but the four-layer devices are not considered in Fig. 1.  Nonetheless, the description of the four-layer structure starts at the beginning of the Theoretical Background section.  The way of description will confuse the readers.

Authors' response: New Figure 1: We illustrated a 4-layer device, explained the parameters in the caption and repositioned it in order to be consistent with the text.

6) p 134, VB?

Authors' response: We have corrected, V_BI instead of VB.

7) Strange matrix element in Eq(10)

Authors' response: In general, in the tight binding reprezentation, the Hamiltonian has a block matrix form, more precisely, a block tridiagonal form as in eq. (10). For a single band model the block tridiagonal matrix form of H becomes a simple tridiagonal matrix. We added this description in the manuscript.

8) The coherence of the electron wave function is important for the appearance of resonant structures reflecting the layered geometry. However, the temperature effect is only incorporated into the distribution function of eq(18), and the dephasing of the wave function does not seem to be taken into account.  According to Fig. 5 and Fig. 11, even the lowest energy resonance line is located about 0.7 eV above the Fermi surface (3.71 eV - 3.0 eV).  In terms of temperature, this is an extremely high value exceeding 8000K. No matter how thin the barrier layer is, the reader may wonder how electrons with such high temperature can maintain sufficient coherence.  I think a reasonable explanation is necessary.

Authors' response: Thomas-Fermi model of metals is a non-interacting model in which the electron-electron interaction is accounted for by the Thomas-Fermi dielectric function. Bearing in mind that temperature is a statistical quantity, at a given temperature there is a (slim) chance that some electrons might have an energy of 0.7 eV above Fermi level. We assumed a ballistic transport across the barier, however, to introduce the decoherence with respect to tunneling, the electron-phonon interaction should be considered. This interaction might be important, especially the electron-optical phonon interaction, which is supposed to be large due to the large dielectric constant. Such a study may be the subject of a forthcoming paper. On the other hand, the electron-electron interaction is not important in our model since there is no charge accumulation in the single-barrier tunneling, like in more complex structures, for ex. in double-barrier structures [Bowen et al., ¨Quantitative simulation of a resonant tunneling diode¨, J Appl. Phys. 81, 3207 (1997)].

Submission Date

28 January 2022

Date of this review

01 Mar 2022 14:30:39

Round 2

Reviewer 2 Report

The authors did not round out their discussion based on the consideration of my comments, but it may proceed to publication with the acknowledgement that its scientific readership will consequently be more limited.

Author Response

We thank the reviewer for his/her assessment of the manuscript. We do not have any further comments to add at this stage.